# Role of *acuK* in Control of Iron Acquisition and Gluconeogenesis in *Talaromyces marneffei*

**DOI:** 10.3390/jof7100798

**Published:** 2021-09-24

**Authors:** Artid Amsri, Juthatip Jeenkeawpieam, Panwarit Sukantamala, Monsicha Pongpom

**Affiliations:** Department of Microbiology, Faculty of Medicine, Chiang Mai University, Chiang Mai 50200, Thailand; artid_a@cmu.ac.th (A.A.); Juthatip_jee@cmu.ac.th (J.J.); jedsada_suk@cmu.ac.th (P.S.)

**Keywords:** *acuK*, AcuK, iron, gluconeogenesis, *Talaromyces marneffei*

## Abstract

*Talaromyces marneffei* is a dimorphic pathogenic fungus causing opportunistic infection in immunocompromised patients. It is a facultative intracellular pathogen and is usually found inside the host macrophages during infection. Alternative carbons and iron are the important nutrients associated with intracellular survival and pathogenesis of *T. marneffei*. This study reported the importance of the transcription factor AcuK in control of gluconeogenesis and iron acquisition in *T. marneffei*. Deletion of *acuK* gene in *T. marneffei* resulted in retardation of growth and germination in both mold and yeast phases. Microscopically, Δ*acuK* showed double nuclei hyphae. However, the yeast cells showed normal morphology. The Δ*acuK* failed to grow in iron-limiting conditions. Additionally, it could not grow in a medium containing gluconeogenic carbon sources. Moreover, Δ*acuK* showed higher susceptibility to macrophage killing than the wild type. These results demonstrated that AcuK controlled both iron acquisition and gluconeogenesis, and it could contribute to the pathogenicity of this fungus.

## 1. Introduction

*Talaromyces marneffei* is a dimorphic pathogenic fungus causing systemic opportunistic infection in Southeast Asia. In Thailand, this infection is common in AIDS patients [1]. Infection due to *T. marneffei* has been reported primarily among immunocompromised people in endemic regions and travelers [2]. As a dimorphic fungus, *T. marneffei* can grow either in yeast or mycelial forms, depending on the growing temperatures. In vitro cultivation on Sabauraud’s agar at 25–30 °C grows as the mycelial phase, producing a green velvety colony with red diffusible pigment. At 37 °C, the yeast growth cycle starts with the production of elongated hyphae and then breaks into the arthroconidia-like yeast cells [3]. During natural infection, the engulfed conidia initially change to yeasts to establish the infection. The pathogen is then divided by fission inside the phagosome of macrophages. Multiplication inside the phagosome environment needs sufficient nutrients. Therefore, the nutrient acquisition process is one of the important mechanisms for fungal pathogenesis [4]. Currently, the intracellular survival mechanism in *T. marneffei* has not been well-elucidated. This study focused on the coupling transcriptional control of iron acquisition and gluconeogenesis as it could help to explain how the fungus act in nutrient-starvation conditions. 

Because of an immune response to the intracellular pathogen infection, the phagosome condition is changed from normal stage to acidic, oxidant-enriched condition. Most nutrients including carbon, nitrogen, and elements are depleted from the host’s nutritional immunity [4]. Iron is one of the essential elements for cellular metabolisms in both humans and *T. marneffei.* The concentration of irons inside the phagosome compartment is usually limited for the growth of pathogens. Therefore, the tug-of-war to compete for the irons presumably happens during infection [5,6,7,8]. *T. marneffei* has developed at least two mechanisms for iron retrieval: reductive iron assimilation (RIA) and siderophore-assisted iron acquisition [9]. However, the control for iron acquisition has not been well-characterized. Besides, the carbon sources that are necessary for fungal growth are also limited in the phagolysosomes [10]. Therefore, the fungus must reprogram their carbon metabolisms to use gluconeogenic substrates as the alternative carbon sources [4]. Gluconeogenesis is a likely mechanism to be used in *T. marneffei* as the alternative pathway to produce glucose from the gluconeogenic substrates [11]. 

AcuK is a transcription factor categorized in the Zn(2)Cys(6) motif group. It was described to control the gluconeogenesis pathway in *Aspergillus nidulans* [12,13]. AcuK could also control the siderophore biosynthesis pathway in *Aspergillus fumigatus.* The deletion of *acuK* caused the halt of growth in iron-depleted conditions [14]. Interestingly, although *A. nidulans* and *A. fumigatus* belong to the same genus, the AcuK function is diverse. We, therefore, questioned how the AcuK plays roles in the dimorphic fungus *T. marneffei.* Recently, the *T. marneffei acuK* transcript was demonstrated to be elevated in both iron-depleted conditions and growth on gluconeogenic substrates [15]. Therefore, we assumed that this transcription factor could control iron and carbon metabolism in this fungus, in a similar manner as previously shown in *A. fumigatus*. To prove this hypothesis, the *acuK* gene was subjected to targeted gene deletion and phenotypic characterization. The ability to use iron and gluconeogenic carbon sources was observed. Additionally, the virulence contribution of *acuK* has been shown in the macrophage infection model. 

## 2. Materials and Methods

### 2.1. Fungal Strains and Culture Conditions 

*Talaromyces marneffei* wild-type ATCC18224 (FRR2161) and uracil auxotroph G816 strain (Δ*ligD niaD^−^ pyrG*^−^) were used in this study [16,17]. *T. marneffei* G816 strain was maintained on *Aspergillus* minimal medium (ANM) supplemented with 5 mM uracil and 10 mM (NH_4_)_2_SO_4_. *T. marneffei* FRR2161 wild-type strain (ATCC18224), ∆*acuK* mutant strain (∆*acuK*, *pyrG^+^*) and complemented strain (∆*acuK*::*acuK pyrG^+^*) were cultured on the ANM without uracil. The conidia were collected from a 10-day-old culture by scraping the colony surface and resuspending them in a sterile normal saline-tween solution (0.1% *v*/*v* tween 40, 0.85% *w*/*v* NaCl). The suspension was filtered through sterile glass wool to remove the mycelia. The conidia concentration was enumerated by counting with a hemacytometer.

### 2.2. Talaromyces marneffei acuK Gene Deletion and Complementation

The *acuK* open reading frame plus 1.5 kb flanking regions were amplified from the genomic DNA of FRR2161 strain with primers 5′*acuK*-*Not*I-F (5′-TAAGCGGCCGCCCGT TGACCGTTGAAGTTGAC-3′) and 3′*acuK*-*Sac*I-R (5′-TAAGAGCTCTCAGGAGATTGA GGTGCGTTG-3′). The 5.083 kb product was cloned into a pGEM-T easy (Promega), generating an *acuK* cloning plasmid, pAA_*acuK*. Then, the pAA_*acuK* was digested with *Sma*I and *Eco*RV to remove 629 base pairs containing a part of the *acuK* open reading frame. The cut plasmid (7.414 kb) containing the residual *acuK* gene was purified and ligated with an *Aspergillus nidulans pyrG* (*AnpyrG*) selectable marker, which was cut from a pAB4626 plasmid (provided by Andrianopoulos, University of Melbourne, Australia) to generate a deletion plasmid pAD_*acuK*. The deletion construct was amplified with primers 5′*acuK*-F1 (5′-CCGTTGACCGTTGAAGTTGACTACG-3′) and 3′*acuK*-R1 (5′-TCAGGAGATTGAG GTGCGTTG-3′) to generate a linear transformed DNA. Transformation of the deletion cassette to the G816 protoplast was performed by using a PEG/CaCl_2_ method as described previously [17]. 

To generate the complemented construct, the *acuK* gene plus promoter region was amplified using *Sac*I-*acuK*_F (5′-GAGCTCCATTATCATGTGTACGAGCCGG-3′) and *Sac*I-*acuK*_R (5′-GAGCTCAACCACCACGAGTTAACACGG-3′) primers and ligated into a dephosphorylated *Sac*I-digested *pyrG*-targeting plasmid (pLS7408) [16], yielding the pAC_*acuK* complemented plasmid. For recycle use of the selectable marker, the *acuK* mutant was induced with 5-fluoroorotic acid (5-FOA) and a survived colony was used to transform it with one microgram of the pAC_*acuK*.

### 2.3. Macroscopic and Microscopic Morphology Examination

The conidia suspensions of wild-type ATCC18224 (FRR2161), Δ*acuK* mutant and complemented strains were prepared at concentrations of 10^3^–10^7^ conidia/mL. Five microliters of each dilution containing 5 to 5 × 10^4^ conidia was dropped on to the surface of the ANM medium. Colony morphology and diameters were observed after cultivation at 25 °C and 37 °C for 12 days. 

For microscopic morphology examination, the agar-coated slide culture was performed according to the method described previously [17]. The cultures were incubated at either 25 °C or 37 °C for 3 days and 5 days, respectively. The fungal cells on slides were stained with a 0.1 mg/mL calcofluor white and a 1 μg/mL 4′,6-diamidino-2-phenylindole (DAPI). The microscopic morphology was visualized under a fluorescence microscope at 460–488 nm (Nikon Eclipse 50i, Tokyo, Japan). 

### 2.4. Germination Assay

The 10^8^ conidia/mL of *T. marneffei* FRR2161, Δ*acuK* and complemented strains were inoculated into Sabouraud’s dextrose broth and incubated at either 25 °C or 37 °C with continuous shaking at 150 rpm to provide the mold and yeast growth. The conidial germination was observed under a light microscope every 3 h for up to 24 h. The percentage of germination was determined by counting the germinated conidia in a total of 1000 inoculated conidia. The experiments were performed in three biological replicates.

### 2.5. Determination of Growth in Iron-Depletion and Iron-Repletion Conditions

To study the effect of iron on the growth of *T. marneffei*, 10^8^ conidia/mL of each strain were prepared in the ANM broth with different iron conditions: normal (ANM, 7 µM ferrous sulfate), iron depletion (ANM, 100 µM phenanthroline) and iron repletion (ANM, 100 µM phenanthroline and 1 mM FeCl_3_). They were cultured at either 25 °C for 36 h or 37 °C for 60 h. The fungal cells were collected by 0.45 mM membrane filtration and dried at 50 °C for 5 days. The growth was determined by the measurement of the dry weight. 

To determine the growth on solid medium, 5 µL of the conidial suspension containing 5 to 5 × 10^4^ conidia was dropped onto the surface of ANM medium (normal, iron-depletion and -repletion conditions) and then incubated at either 25 °C or 37 °C. The colony was observed after incubation for 7 days.

To study the expression profiling of *T. marneffei*, conidia of each strain was prepared in the ANM broth to a final concentration of 10^8^ conidia/mL. The culture was grown at 25 °C for 16 h. Then the cells were harvested by centrifugation and transferred into an iron-free ANM broth (ANM supplemented with trace element solution without FeSO_4_) with the addition of FeCl_3_ to different iron final concentrations: 10 µM and 1000 µM for low and high iron conditions, respectively. After that, the culture was continued for an additional 24 h at 25 °C before we harvested the cells for RNA extraction. 

### 2.6. Determination of Growth on Medium Containing Gluconeogenic Substrates

A carbon-free agar medium, ANM without glucose, was prepared. The gluconeogenic carbon sources were added as following: 50 mM proline, 50 mM acetate and 0.5% ethanol. A 1% glucose was added in the carbon-free medium to serve as a growth control. The 10^3^–10^7^ conidia from *T. marneffei* FRR2161, Δ*acuK* and complemented strains were spotted onto the surface of the agar medium and incubated at either 25 °C or 37 °C. The colony was observed after incubation for 7 days.

### 2.7. Real-Time Reverse Transcription-Polymerase Chain Reaction (Real-Time RT-PCR) 

The 60 h mycelium and 96 h yeast cultures were harvested by centrifugation at 4500 rpm for 10 min. Approximately 2 g wet weight was homogenized in a TRIzol reagent (Invitrogen, Life Technologies, Carlsbad, CA, USA) by using a bead beater (Biospec, Bartlesville, OK, USA). The total RNA was isolated following the manufacturer’s protocol. The RNA was digested with 1 U/µL DNase I to eliminate the residual DNA and purified by isopropanol precipitation. RNA concentrations were determined with a spectrophotometer (Nanodrop 2000: Thermo Scientific, Waltham, MA, USA). Two micrograms of the total RNA was converted into complementary DNA by using a ReverTra Ace^®^ qPCR RT Master Mix (TOYOBO, Osaka, Japan). 

Real-time RT-PCR on the desired transcript was performed by using the SYRB Green qPCR mix (Thunderbird SYBR Green Chemistry, TOYOBO), and the fluorescent signal was detected in a 7500 Real-Time PCR System (Applied Biosystems, Foster City, CA, USA). Primers used for amplification of *acuK* transcript are Real_*acuK*_F primer (5′-CCTCCGCCACGGATCATAGTG-3′) and Real_*acuK*_R primer (5′-ACACCGTCGTGGCATGCATC-3′). Primers used for an *abaA* transcript are Real_*abaA*_F (5′-TGGAGGAGTA GGAGGAGGTG-3′) and Real_*abaA*_R (5′- GATGGAACGGAACAGGAGCA-3′). The real-time PCR condition was 1 cycle of 95 °C for 60 s, followed by 40 cycles of 95 °C for 60 s. An actin gene was used as endogenous control (primer sequences; Act1F, TGATGAGGCACAGTCTAAGC and Act1R, CTTCTCTCTGTTGGACTTGG). Calculation of a relative expression was performed using formula 2^−(ΔC^_t_^)^ (where ΔC_t_ = C_t actin_ − C_t target_).

### 2.8. Chrome Azurol S (CAS) Assay

To measure the total siderophore activity of each strain, the chrome azurol S (CAS) colorimetric method was used as described previously [18]. Briefly, 1 × 10^6^ conidia of each fungal strain were inoculated into ANM broth and culture at 25 °C for 2 days. The cells and culture supernatant were harvested after centrifugation at 10,000 rpm for 20 min. The cells were mechanically broken with 0.1 mm glass beads in a bead beater (Biospec, Bartlesville, OK, USA). Total protein concentration was determined in cell lysate and culture supernatant and then adjusted to 2 mg/mL. The siderophore activity in 2 mg of total protein was measured by adding 0.15 mM chrome azurol S, 0.015 mM FeCl_3_, 1.5 mM hexadecyl trimethyl ammonium bromide (HDTMA) and 1 M piperazine (pH 5.6), determining the absorbance at 630 nm. An EDTA (Sigma-Aldrich, St. Louis, MO, USA) was used to generate a standard curve. Using this standard curve, the siderophore production of each fungal strain was calculated in terms of micromoles per gram of total protein.

### 2.9. Macrophage Infection for Phagocytosis and Killing Assay

A THP-1 human monocytic cell line is used for the macrophage infection model. The cells were maintained in RPMI 1640 medium (Life Technologies, Grand Island, USA) with 10% FBS (*v*/*v*) at 37 °C, 5% CO_2_. The THP-1 was seeded at a concentration of 1 × 10^6^ cells per well into a 6-well plate containing one sterile coverslip for phagocytosis assay and a 12-well plate for killing assay. The cell was activated with 32 μM phorbol 12-myristate 13-acetate (PMA) (Sigma-Aldrich) for 24 h at 37 °C and 5% CO_2_. Then, 1 × 10^6^
*T. marneffei* conidia were added to the macrophages and infected for 2 h. For the phagocytosis assay, macrophages were fixed in 4% paraformaldehyde and stained with 1 mg/mL fluorescent brightener 28 (calcofluor-CAL) to observe the fungal cell walls. Mounted coverslips were examined using differential interference contrast (DIC) and epifluorescence optics for cell wall staining and viewed on a Reichart Jung Polyvar II microscope. Images were captured using a SPOT CCD camera (Diagnostic Instruments Inc., Sterling Heights, MI, USA). The number of phagocytosed cells was recorded in a population of approximately 100 macrophages in three independent experiments. The phagocytic index (the number of phagocytosed conidia per macrophage) was determined by dividing the total number of intracellular conidia by the number of macrophage cells containing conidia. For the killing assay, non-phagocytosed/non-adhered conidia were washed three times with PBS. Macrophages were lysed with 0.25% TritonX-100 (Sigma-Aldrich) and fungal cells were recovered and plated on a synthetic dextrose (SD) agar (0.17% (*w*/*v*)) yeast nitrogen base without ammonium sulfate and amino acids (2% (*w*/*v*) glucose, 10 mM (NH_4_)_2_SO_4_, 2% agar). CFUs were determined after growth at 37 °C for 7–10 days. Data were expressed as the mean value deviation from triplicate measurements.

### 2.10. Statistical Analysis 

The data were analyzed, depending on the experiment, with a Student’s *t*-test and Tukey’s multiple comparison test or unpaired *t*-test and Welch’s correction with a significant value of *p* < 0.05. All statistical analysis was performed by using SPSS v. 16.0 and Prism software (GraphPad, version 7.0).

## 3. Results

### 3.1. Mutation of acuK Reduced Growth and Conidial Germination 

Colony morphology was compared on *T. marneffei* FRR2161 wild-type (WT), Δ*acuK* and complemented strains (Δ*acuK*::*acuK*). For the mycelial phase, Δ*acuK* grew slower than the wild-type and complemented strains; therefore, it showed delay conidiation but with normal appearance and pigmentation (Figure 1A,B). The germination rate of the conidia was determined microscopically every 3 h after conidial inoculation into ANM broth for up to 24 h. The germination of Δ*acuK* started at the same time as the wild-type and complemented strains; however, the number of conidia that could germinate was significantly lower. In the wild-type strain, 40%, 85% and 100% of conidia germinated 9, 12 and 15 h after incubation, respectively, and close percentages were observed in the complemented strain. There was a significant difference in the germination rates between wild-type and Δ*acuK* strains (Figure 1C). This result indicated that the absence of the *acuK* resulted in slow growth and reduced viability of the conidia. 

A similar result was observed in the cultivation at 37 °C. The colony diameter of Δ*acuK* was smaller than the wild-type and complemented strains (Figure 1D,E). The germination at 37 °C normally starts slower than 25 °C. The Δ*acuK* showed a severe reduction in the percentage of germinated conidia, as shown in Figure 1F. Altogether, these results revealed that mutation of the *acuK* gene caused defects in the growth and survival of conidia in both mold and yeast phases. 

### 3.2. ΔacuK Showed Abnormal Hyphae with Double Nuclei

Microscopic visualization of *T*. *marneffei* strains was performed on both mold and yeast phase cultures on ANM agar-coated slides after staining with calcofluor white and DAPI. Interestingly, while the yeast cells of Δ*acuK* showed a normal, one cell-one nucleus morphotype (Figure 2B), the hyphal compartment showed abnormal double nuclei (Figure 2A). This result implied that the deletion of *acuK* could have resulted in the cell division defect. The molecular mechanism in control of cell division in fungi is complex. The underlying reason for this defect has yet to be uncovered in *T. marneffei*. A previous study demonstrated a similar phenotypic change in *T. marneffei abaA* mutant strain [19]. AbaA is a transcription factor related to morphogenesis. Overexpression of *abaA* resulted in the multinucleate vegetative filamentous cells. Therefore, we analyzed the *abaA* transcript level in Δ*acuK* and found that it was two times upregulated compared to the wild-type strain (Figure 3). The result suggested that AcuK may control morphogenesis in *T. marneffei* via the AbaA transcription factor.

### 3.3. AcuK Plays a Role in Iron Assimilation

To test whether *acuK* is involved in iron acquisition *T. marneffei*, the wild-type, Δ*acuK*, and complemented strains were cultivated in regular ANM medium, defined as normal condition (containing 7 µM of irons), and iron-depletion (ANM with 100 µM phenanthroline) and -repletion (ANM with 100 µM phenanthroline and 1 mM FeCl_3_) conditions. Tests were performed at either mold or yeast phases. At 25 °C, the mold colony morphology of all strains was normal in normal and iron-repletion conditions. They produced a velvety to fluffy colony with green color when aged. The high amount of iron in the repletion condition sped up the growth in comparison to the normal medium, indicating that high iron could enhance the growth of the fungus. Severe growth retardation was observed in Δ*acuK* in the iron-depleted condition (Figure 4A). Growth was also confirmed by the determination of dry weight and the result corresponded to the growth on solid ANM medium. A significant reduction in hyphal mass was found in Δ*acuK* (Figure 4B). 

At 37 °C, Δ*acuK* gained severe defect than that was observed at 25 °C. Contrary to the mold phase, the wild-type and complemented strains were unable to grow in the iron-depletion condition. This result indicated that the yeast growth requires higher irons to grow than the mold phase. This evidence was also described previously [9]. The yeast growth of Δ*acuK* was completely inhibited in iron-depleted conditions. However, the yeast-like colony appearance of the mutant was not different from the wild-type and complemented strains when grew on a normal and iron-repleted medium (Figure 4C). The dry weight assay showed a significant lowering in mass (Figure 4D). The corresponded observation was marked as the outcome of *acuK* mutation resulted in the inability of the fungus to grow in the iron-insufficient conditions. This result implied the importance of AcuK in control of iron assimilation. Lack of the *acuK* gene may change the ability to acquire irons in this fungus. The Δ*acuK* was tested in the CAS assay to determine the siderophores content. Amount of total and extracellular siderophores were detected in the cells and culture supernatants, respectively. As shown in Figure 4E, the siderophores content was only slightly decreased in Δ*acuK*. This partial reduction of siderophores suggested that the biosynthesis may be additionally controlled by other factors. 

The siderophore biosynthetic pathway has been described in *T. marneffei* [9]. As the control of siderophore synthesis involved the transcriptional factors SreA and HapX that work in a negative feedback loop [20,21]. We, therefore, used a real-time RT-PCR to detect the transcripts of *sreA* and *hapX* in the Δ*acuK*. When grew the wild-type strain in a low iron condition (10 µM), *sreA* transcript was very low and *hapX* transcript was raised as expected (Figure 5A). The result showed a normal pattern in the wild-type strain in which the *hapX* expression was downregulated to decrease the uptake of iron into the cells (from 0.9 in low iron concentration to 0.13 in high iron concentration) (Figure 5A). Interestingly, the Δ*acuK* showed prominently high *sidA* and *hapX* transcripts at the low iron condition when compared to the level in the wild-type strain (Figure 5B). This result suggested that *acuK* might controlled iron homeostasis via inhibition of *sreA* expression in low iron concentrations. 

Iron assimilation in *T. marneffei* requires both siderophore-assisted iron uptake and reduction iron assimilation (RIA) [9]. To determine the role of AcuK in the control of iron acquisition, we detected the transcript of genes in siderophore production, *sidA* and *sidX*, as well as RIA, *ftrA* and *fetC*. The *sidA* and *sidX* genes encode the enzyme ornithine N^5^-oxygenase, which is important to change ornithine to hydroxyornithine in an initial step of the siderophore biosynthesis pathway. In *T. marneffei*, SidA and SidX were reported to have a different role in the production of different types of siderophores and fungal morphotypes. While SidA plays a role in extracellular siderophore production in the mycelial phase, SidX takes an important role in both intracellular and extracellular siderophore production in the yeast phase [9]. However, the transcriptional control of *sidA* and *sidX* has never been reported. Our result found that *sidA* and *sidX* were differentially expressed. In the wild-type strain under low iron conditions where the *hapX* is high, the *sidX* transcript was higher than the *sidA* transcript (Figure 5C). This result suggested that the HapX may control the expression of *sidX*. However, when observed in the Δ*acuK*, even in the presence of a high *hapX* transcript level, the *sidX* expression was low (Figure 5B,D). This result demonstrated that the AcuK did not control *sidX* expression. Interestingly, Δ*acuK* showed a significant reduction in *sidX* expression at low iron conditions when compared to the level in the wild-type strain (2.32 vs. 0.05) (Figure 5C,D). This result implied that AcuK controlled the expression of *sidX* but not *sidA*. In conclusion, our result found that the control of *sidX* expression involved *acuK* but not *hapX*. As the siderophore biosynthesis and its control have been admitted as complex processes, additional experiments such as overexpression or chromatin immunoprecipitation assay (ChIP) should be performed to unravel these complex mechanisms. For the investigation of *ftrA* and *fetC* expression, the wild type showed a typical expression pattern where they were increased in the low irons and declined in high irons. Our finding found that only *fetC* expression pattern was affected by the mutation of *acuK*. The *fetC* transcript in Δ*acuK* was significantly increased 5.5 and 35 times compared to the wild type at low and high iron conditions, respectively (Figure 5E,F). This result suggested that AcuK could be a negative regulator of *fetC*, the gene encoding an RIA enzyme in *T. marneffei*. 

Altogether, this study found that Δ*acuK* showed a growth defect in iron-insufficient conditions since it produced a fewer amount of siderophores. The underlying mechanism in control of siderophore synthesis is not fully understood, but it could be because the lack of *acuK* resulted in the decrease of *sidA* and *sidX*, which encode the enzymes in the initial step of siderophore biosynthesis. Surprisingly, the mutation of *acuK* caused the upregulation of *sreA* and *hapX* without promoting the siderophore biosynthesis. Additionally, *acuK* may play an important role in the negative control of RIA. The lack of *acuK* resulted in an enhanced *fetC* transcription, probably to compensate for the defect of siderophore production. 

### 3.4. AcuK Is Required for Gluconeogenesis 

The AcuK transcription factor has been reported to involve in the control of gluconeogenesis in both *Aspergillus nidulans* and *Aspergillus fumigatus* [12,13,14]. To test whether *T. marneffei acuK* is involved in gluconeogenesis, the wild-type, Δ*acuK* and complemented strains were cultured on a medium containing gluconeogenic substrates, acetate, ethanol and proline. Both the mycelial and yeast phases of the *acuK* mutant could not grow on all tested media, indicating *acuK* had a role in gluconeogenesis as well as that previously reported in the *Aspergilli* (Figure 6). 

### 3.5. AcuK Mutation Increased Percentages of Phagocytosis and Killing by a Human Macrophage

The differentiated THP-1 macrophage was used as a model study in *T. marneffei* infection. The conidia from wild-type and Δ*acuK* strains were prepared and used in phagocytosis and killing assay. Figure 7 showed the engulfed conidia at the investigated time points 2 and 24 h after infection. The mutant showed the delayed in germination at 24 h. In addition, phagocytosis, the phagocytic index and the percentage of killing were determined at 2 h of infection. As shown in Table 1, the phagocytic index, as well as the percentage of killing, were slightly increased in Δ*acuK*. This result implied that lack of *acuK* may attenuate the cell division ability in vivo.

## 4. Discussion

The *Talaromyces marneffei acuK* gene encodes a C6 finger domain transcription factor. This protein has been reported as a fungal-specific transcription factor and found only in Ascomycetes filamentous fungi. In this study, the function of *acuK* in *T. marneffei* was characterized. AcuK is required for gluconeogenesis and iron acquisition in *T. marneffei* as well as in *A. fumigatus*. However, the deletion of *acuK* in *T. marneffei* also resulted in a defect in growth, conidial viability, growth in iron-depleted conditions and gluconeogenic substrates. 

In a phenotypic study, the wild-type, ∆*acuK* and complemented strains were compared at both macroscopic and microscopic morphology. At 25 °C, the *T. marneffei* ∆*acuK* colony showed slower growth and conidiation than in the wild-type and complemented strains. Similarly, at 37 °C, the mutant displayed no growth in the iron-depleted conditions. Microscopic observation by fluorescent staining with calcofluor white (CFW) and 4′,6-Diamidine-2-phenylindole dihydrochloride (DAPI) found that the ∆*acuK* strain had cell division defect in the mold phase as shown by the double nuclei per cell. Since this phenotype has been noticed before on the *abaA* overexpression strain of *A. fumigatus*, *T. marneffei* and *Fusarium graminearum* [14,19,22], the level of *aba* transcript in ∆*acuK* was investigated and we found that it was upregulated. How the mutation of the *acuK* gene affected the cell division process through the overexpression of *abaA* is unknown. AbaA is the transcription factor that regulates the conidiation process, and it also acts as a regulator for cell division in filamentous fungi. Overexpression and constitutive expression of *abaA* could therefore result in pleiotropic defects and decreased conidiogenesis. We also found in the germination assay that the ∆*acuK* showed a declined germination rate. This result suggested the decreased viability of the propagules produced by the mutant. One possible explanation for this defect could be the reduction in iron uptake and storage. During conidial germination, the cellular enzymes need irons as the cofactor; therefore, they could not properly function and affected the overall metabolism.

The *acuK* function is required in growth during the iron-limited conditions. Similarly, in *A. fumigatus*, the ∆*acuK* mutant could not grow under iron-limited conditions. The mutation of *A. fumigatus acuK* reduced extracellular siderophore production and iron uptake [14]. In this study, the CAS assay found a decreased siderophores level in the *acuK* mutant strain of *T. marneffei*. To understand the involvement of AcuK in the control of siderophore biosynthesis, the levels of expression of the *sreA*, *hapX*, *sidA*, *sidX* were measured in the mycelial phase. Interestingly, both sreA and *hapX* transcript levels were increased in ∆*acuK* despite the fact that the siderophore production was decreased. This result differed from the regular pattern found in *A. fumigatus*. However, even in the presence of a high *hapX* transcript level, the *sidA* and *sidX* transcripts involving in the initial step of siderophore biosynthesis were very low. This evidence suggested that *sidA* and *sidX* may be controlled by AcuK, and the detailed mechanism is yet to be determined. Our finding also indicated, for the first time, that *acuK* could control the RIA system via the inhibition of *fetC* expression. Normally, FetC and FtrA form complexes that function in the oxidation of ferric (Fe^3+^) to ferrous (Fe^2+^) and transportation into the fungal cell. 

The *acuK* function is required for gluconeogenesis in *T. marneffei*. Deletion of the *acuK* made the fungus not grow on the medium containing acetate, proline and ethanol gluconeogenic substrates, which were used as the alternative carbon sources. The mechanism for control of gluconeogenesis was described in *A. nidulans*. AcuK works with its partner transcription factor AcuM as a heterodimer in control of target genes in gluconeogenesis pathways. They are co-regulated in the control of genes involving in gluconeogenesis but not iron metabolism [13,23]. In *A. nidulans*, the *acuK* mutant loss of induction of the *acuF*, *acuN* and *acuG* genes, encoding enzymes in the TCA cycle, intermediates production [13]. Whether *T. marneffei* AcuK controls the same set of enzymes or works together with AcuM is needed to be verified in the future.

Our findings that *acuK* in *T. marneffei* governs both iron acquisition and gluconeogenesis found the same role that was previously described in *A. fumigatus*. The interesting point is that they both are human pathogenic fungi. Iron acquisition is one of the virulence factors in *A. fumigatus*, and lacking siderophore synthesis led to attenuation of the virulence [14,15]. It is highly feasible that iron is an important virulence factor for *T. marneffei* as well, since the growth of this fungus depends on iron availability [7]. Whether the deletion of the *acuK* gene affects the virulence in *T. marneffei* as it does for *A. fumigatus* remains to be elucidated by the mouse infection model. Together, nutrient acquisition is the virulence-assisted process for fungal adaptation and intracellular survival; therefore, targeting AcuK could be an alternative therapeutic intervention. This study demonstrated that AcuK contributes the pathogenesis of *T. marneffei*, and most importantly, it is uniquely found only among Ascomycetous fungi [15]. The absence of AcuK transcription factor in mammalian cells makes it an attractive target for the development of selective antifungal agents targeting the central metabolism. Currently, development of therapeutics using transcription factors as the drug targets have been increasingly studied. Inhibition of the transcription factor regulatory function could be done by inhibition of binding, either between transcription factor subunits or the transcription factor and DNA-binding sites [24,25]. Discovery for molecules such as monoclonal antibodies or antifungal peptides (AFP), which could impede AcuK binding to other cooperate transcription factors or the DNA-binding domain (DBD), could be useful to prevent the activation of under-controlled target gene expression.

## 5. Conclusions

The mutation of *acuK* in *T. marneffei* caused the phenotypic defects in ability to grow in iron-depleted conditions and gluconeogenic substrates. The mutant showed delayed growth, conidiation and double nuclei in the mycelial phase. The mutant could not grow as hyphae inside the macrophage. Moreover, the susceptibility to the macrophage phagocytosis and killing was increased in the mutant. 

The *acuK* governs both iron acquisition and gluconeogenesis, the same role that was previously investigated in *A. fumigatus*. However, the control of iron assimilation is different. The control of *sidA* and *sidX* expression seems to be more involved to *acuK*, not *hapX*. In addition, *acuK* could negatively control the *fetC* transcription and the RIA system in *T. marneffei*. As iron assimilation is the complex process, additional studies are needed to fully understand its control mechanism. 

## Figures and Tables

**Figure 1 jof-07-00798-f001:**
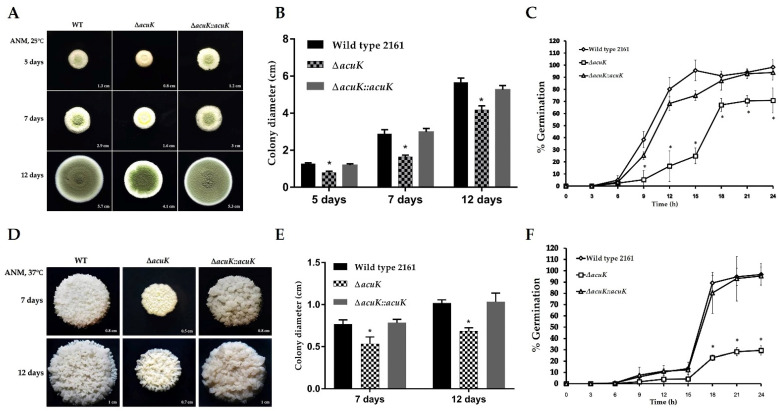
Colony growth and conidial germination of *T. marneffei* strains. Conidia from wild-type FRR2161, Δ*acuK* and complemented strains were inoculated on ANM agar and incubated to grow at 25 °C and 37 °C. At 25 °C, the mold colony was observed at 5, 7 and 12 days (**A**). The colony diameters were measured (**B**). The percentage of germination was determined (**C**). At 37 °C, the yeast colony was observed at 7 and 12 days (**D**). Yeast-like colony diameters were measured (**E**) and the percent of germination at 37 °C was calculated (**F**). Asterisks (*) indicate data that significantly differed (*p* < 0.05) based on Tukey’s test.

**Figure 2 jof-07-00798-f002:**
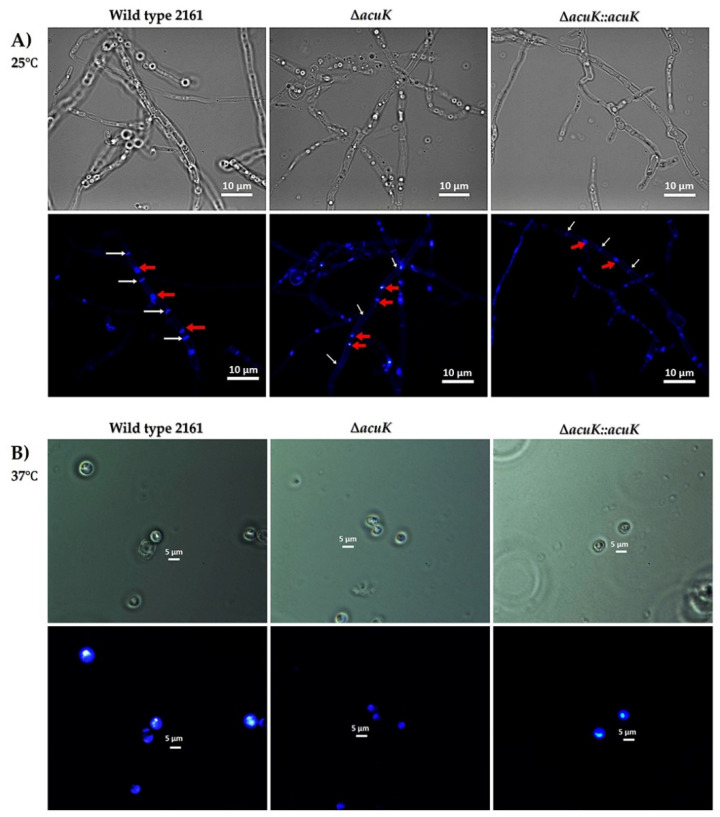
Microscopic morphology of hyphae and yeast cells of Δ*acuK.* Agar-coated slides of *T. marneffei* 3-day mold culture and 5-day yeast culture were stained with calcofluor white and DAPI. Hyphal cells with double nuclei are shown indicated by red arrows. The white arrows indicated septum (**A**). The yeast cell shows normal morphology with one nucleus per yeast cell (**B**).

**Figure 3 jof-07-00798-f003:**
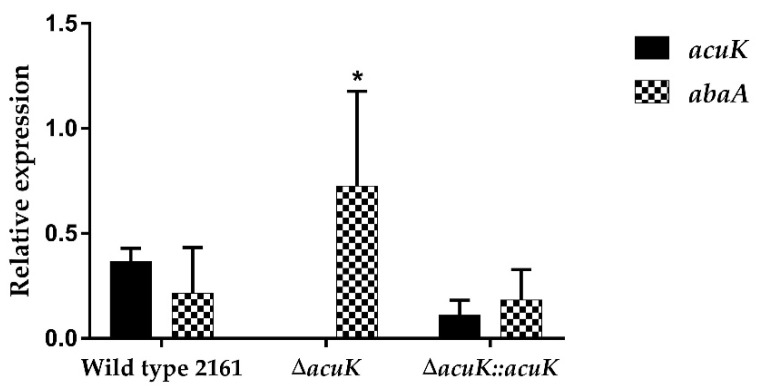
Expression of *abaA* transcript in Δ*acuK*. The transcript levels were analyzed by real-time PCR in the wild-type, Δ*acuK* and complemented strains and normalized with actin transcript. Asterisk (*) indicate data with significantly differed at *p* < 0.05.

**Figure 4 jof-07-00798-f004:**
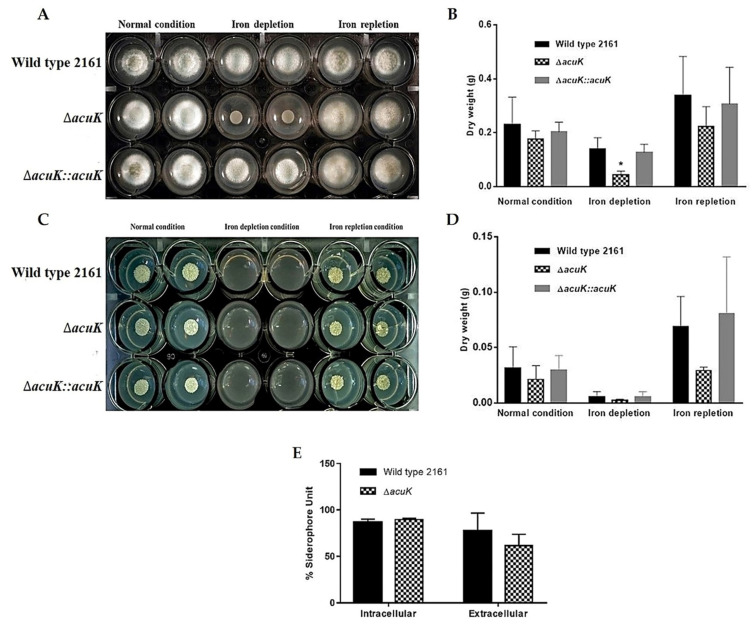
Growth of *T. marneffei* strains in different iron conditions. Cultures were prepared by inoculating the conidia into ANM medium at a final concentration of 10^8^ conidia/mL. The media were prepared for the different iron conditions; normal medium (ANM, 7 µM ferrous sulfate), iron-depletion medium (ANM + 100 µM phenanthroline) and iron-repletion medium (ANM + 100 µM phenanthroline and 1 mM FeCl_3_). They were cultured to the mold phase at 25 °C for 36 h on solid medium (**A**) and liquid medium (**B**). The yeast phase culture was incubated at 37 °C for 60 h on solid medium. Asterisk (*) indicate data with significantly differed at *p* < 0.05. (**C**) and liquid medium (**D**). The siderophore content in the wild-type and Δ*acuK* strains were determined by CAS assay (**E**).

**Figure 5 jof-07-00798-f005:**
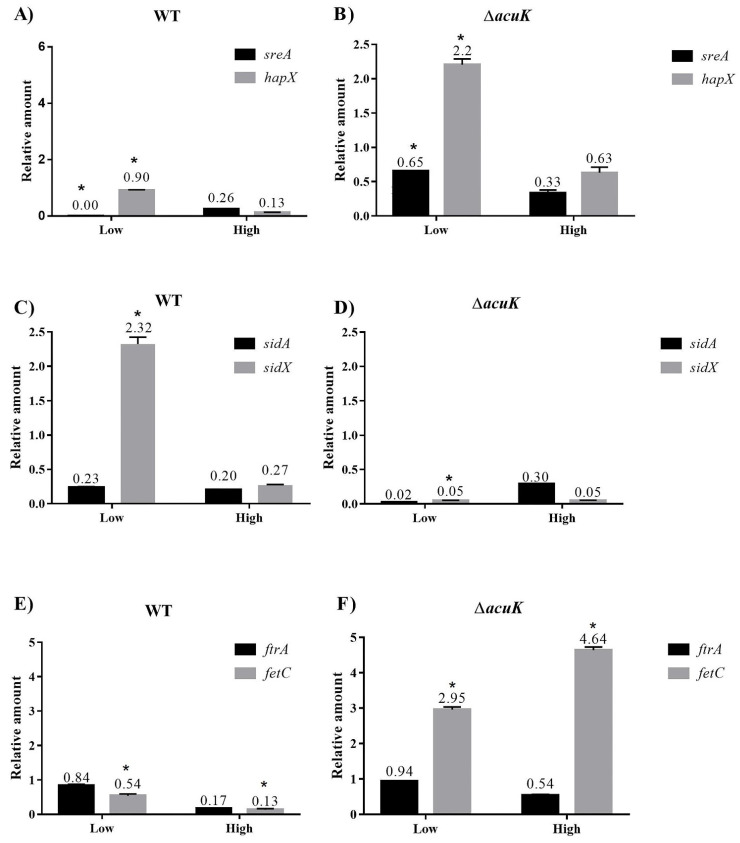
Real-time PCR measurement of transcript levels in siderophore biosynthetic and reductive iron assimilation pathway in the mold phase of Δ*acuK*. RNA levels of these indicated genes were measured and normalized with *actin* transcript in the wild-type and Δ*acuK* strains, respectively; *sreA* and *hapX* (**A**,**B**), *sidA* and *sidX* (**C**,**D**), and *ftrA* and *fetC* (**E**,**F**). The RNA level for each gene was normalized to the internal control actin transcript. The number over the bar indicated the average relative amount of indicated genes (triplicate cultures). The significant level for each gene (*) in the Δ*acuK* was compared to the level in the wild-type strain at the same iron condition as determined by the unpaired test, Welch’s correction (GraphPad Prism, version 7.0).

**Figure 6 jof-07-00798-f006:**
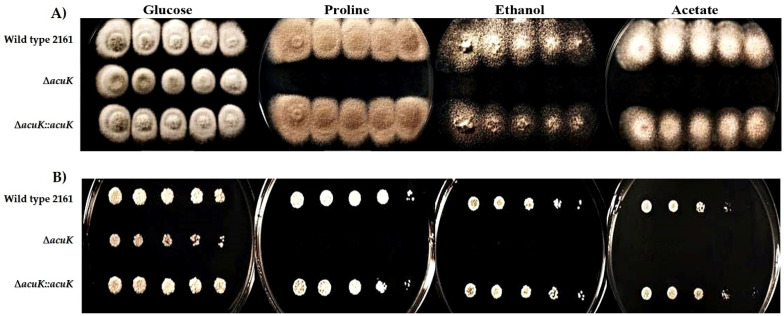
Growth on gluconeogenic substrates containing medium. Conidia suspensions were prepared at a concentration of 10^3^ to 10^7^ conidia/mL. Five microliters of each conidial suspension containing 5 to 50,000 conidia was spotted onto the surface of the medium supplement with glucose, proline, ethanol and acetate. At 25 °C, the mold colony of wild-type FRR2161, Δ*acuK* and complemented strains were observed at 12 days (**A**). At 37 °C, the yeast colony was observed at 12 days (**B**).

**Figure 7 jof-07-00798-f007:**
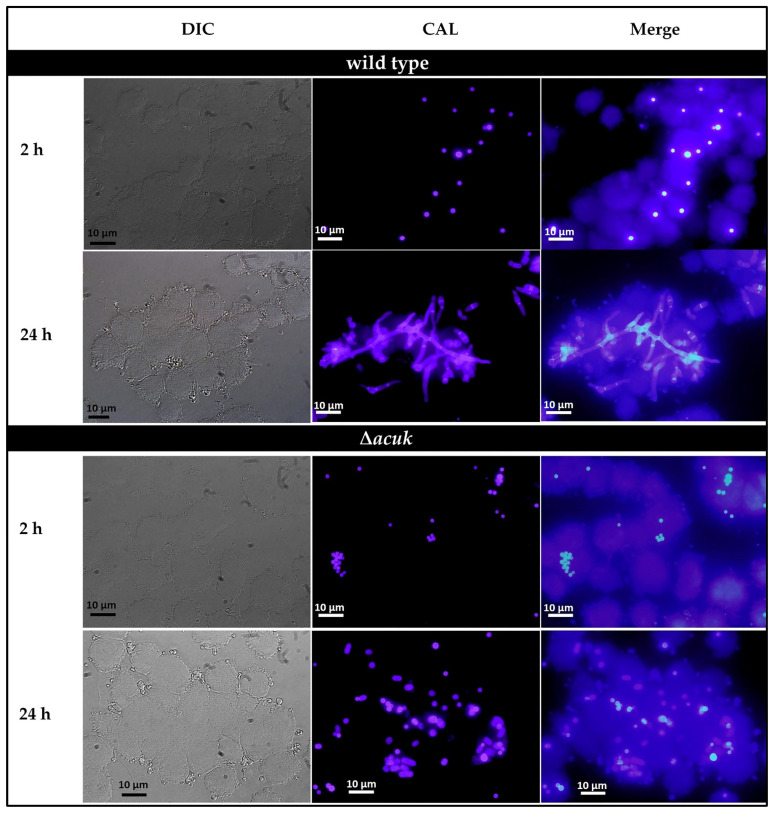
Fluorescence microscopic observations of calcofluor white-labelled *T. marneffei* infected THP-1 human macrophage cells. PMA-activated THP-1 human macrophages were infected with conidia from the wild-type and ∆*acuK* strains for 2 and 24 h before they were examined microscopically. At 24 h post-infection, the wild-type strain showed elongated cells with septation. In contrast, the ∆*acuK* mostly produced only fission yeast cells and some other cells (100X magnification, scale bars are 10 µm).

**Table 1 jof-07-00798-t001:** Phagocytosis and killing assay of *T. marneffei* by differentiated macrophage THP-1 (mean ± SD).

*T. marneffei* Strain	% Phagocytosis	Phagocytic Index (PI)	% Killing
Wild type	40 ± 0.04	1.43 ± 0.01	66.12 ± 1.86
Δ*acuK*	41 ± 0.12	3.39 ± 0.00	73.66 ± 2.80

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
