# Peer review of "Role of *acuK* in Control of Iron Acquisition and Gluconeogenesis in *Talaromyces marneffei"

_jof, 2021, doi:10.3390/jof7100798_

Round 1

Reviewer 1 Report

Line 127: 

Why was phenanthroline added under iron repletion condition?

Figure 4B, 4D:

I was wondering if the expression of siderophore receptors had an effect on the growth inhibition of acuk deletion strain. Did you check that the expression level of the siderophore receptors in the acuK deletion strain is the same as that in WT?

Figure 4A, 4C:

The "w" in the "wild type" word is missing.

Figure 4E:

In Figure 4, It should be shown that there are statistical differences between extracellular siderophore content between acuK deletion strain and WT.

Line 327, Figure 5C:

Since there is no statistical difference in the sidA expression between acuK deletion strain and WT Figure 5C, I think that acuK is not involved in the control of sidA.

Table1:

Please show each data in table 1 as mean ± SD.

Author Response

Reviewer’s comment  Author’s response

Line 127: 

Why was phenanthroline added under iron repletion condition?

Since we did not use iron-free water and chemicals to prepare the culture medium in this study, phenanthroline is designed to be used as the iron chelator to deplete irons in the ANM medium. We only needed the minimum amount of phenanthroline that could inhibited the growth of T. marneffei. Therefore, in our previous study, the phenanthroline concentration was varied to determine the minimum concentration (MIC90) that could inhibit fungal growth and found its optimal concentration at 100 mM (Amsri A, Pongpom M. Effects of iron on acuK expression in Talaromyces marneffei. Proceedings in 2nd National Graduate Research Conference and Creative Innovation Competition. May 17-78 2018, The Empress Hotel, Chiang Mai, Thailand.  Pages 73-80.). Therefore, this concentration can be used to normalize the conditions in both iron-depletion and repletion without additional effect of excess chelation. Presence of the fungal growth in the iron-replete culture represent the effect from the additional iron that we supplied.  

Figure 4B, 4D:

I was wondering if the expression of siderophore receptors had an effect on the growth inhibition of acuK deletion strain. Did you check that the expression level of the siderophore receptors in the acuK deletion strain is the same as that in WT?

Thank you for your question. Unfortunately, we did not check the expression of genes encoding siderophore receptors in this study. We knew (from literature review on studies in A. fumigatus) that the expression of iron receptors or transporters could be varied upon the iron concentrations. However, we cannot deny that it could also been changed in the presence or absence of transcription factor such as AcuK. Therefore, we accept this concept in mind and plan to include the assays in our ongoing study in determination target genes of AcuK.

Figure 4A, 4C:

The "w" in the "wild type" word is missing.

The missing "w" in the "wild type" word was added into Figure 4A and 4C (revised manuscript).

Figure 4E:

In Figure 4, It should be shown that there are statistical differences between extracellular siderophore content between acuK deletion strain and WT.

There was “no significant difference” in extracellular siderophore content between acuK deletion and WT strains. We think that the siderophore production in the acuK mutant could be from the function of another homologous transcription factor, AcuM. This hypothesis came from a study in A. fumigatus that found overlapping function between AcuK and AcuM (Pongpom, M. et al.  Divergent targets of Aspergillus fumigatus AcuK and AcuM transcription factors during growth in vitro versus invasive disease. Infect Immun 2015, 83(3), 923-933). However, this hypothesis is now under investigating.

Line 327, Figure 5C: Since there is no statistical difference in the sidA expression between acuK deletion strain and WT Figure 5C, I think that acuK is not involved in the control of sidA.

 We agreed that the acuK is not involved in the control of sidA. Thus, the "sidA" word was removed from line 327 in the revised manuscript.

Table1:

Please show each data in table 1 as mean ± SD.

The information on SD values were added in the Table 1 (revised manuscript).

Reviewer 2 Report

The study titled as ‘Role of acuK in control of iron acquisition and glucogenesis in Talaromyces marneffei’ has described the functions of the transcription factor AcuK in a pathogenic fungus Talaromyces marneffei via deletion of acuK gene. This study seems to be a follow-up work from the previous study (Pongpom et al. 2020. J. Fungi) which suggested that the Involvement of acuM and acuK in gluconeogenesis and iron homeostasis in T. marneffei via transcript analysis. In the current study, mutation of acuK gene has been clearly shown to be critical for iron acquisition (especially under the iron depletion condition) and gluconeogenesis, which are likely connected to the fungal pathogenicity. Although there are other studies describing same or similar roles of AcuK in the other fungus such as A. fumigatus, the current study still has a merit to be published since it is about different fungal genus and species, and this would contribute to future studies for treatment throughout many other pathogenic fungal species.

The manuscript has been well written and I do not see any major problems to understand the results which are clearly elucidated to support the authors’ ideas. I do not have any comments for the manuscript revision before the possible publication.

Author Response

Thank you very much.  We appreciated your valuable comments.

Reviewer 3 Report

The authors in this research work demonstrated the importance of AcuK in the metabolism and pathogenesis of Teleromyces marneffei. The manuscript is well written with sound research design and presentation. However, there are few minor concerns which needs to be addressed. 

  1. Authors have demonstrated that the deletion of AcuK leads to higher susceptibility against macrophage killing. Another important population of leukocyte which play crucial role in fungal and bacterial killing is Neutrophils. Therefore, it will be ideal if author can generate some relevant data showing the susceptibility against another subset of leukocyte which play key role in dysregulated immune responses and sepsis.
  2. The important finding that the absence of acuK make this fungal susceptible is very interesting but author should discuss the therapeutic scope of these findings in the discussion section. What type of future studies should be conducted to progress this work for the translational aspects of these findings. 

Author Response

Thank you for the valuable comments and suggestions. We would like to respond your suggestions as following;

Suggestions

Response to reviewer

Authors have demonstrated that the deletion of AcuK leads to higher susceptibility against macrophage killing. Another important population of leukocyte which play crucial role in fungal and bacterial killing is Neutrophils. Therefore, it will be ideal if author can generate some relevant data showing the susceptibility against another subset of leukocyte which play key role in dysregulated immune responses and sepsis.

In this article, we selected primarily to use only the macrophage since it is the most powerful tool to study the host-pathogen interaction. In vivo, alveolar macrophage is the first leukocyte population in first line defense mechanism to engulf the fungal conidia. For T. marneffei’s pathogenesis, the macrophage could not kill the fungus effectively and led the fungal growth inside the phagosome. The fungal adaptation is therefore accounted as the process that we are trying to inhibit as we proposed in the discussion section of this manuscript. Currently, we only have a developed macrophage infection model available in our laboratory.

The idea to study the innate immune response especially on neutrophils is excellent. We cannot deny that neutrophils play role in T. marneffei resistance since we found this fungal infection in neutropenic patients. It will be very useful to set up the experiment to see T. marneffei and neutrophil interaction. We have set this idea as one of the aspects to be investigated in our research group and plan to perform the experiment by using primary neutrophil in the near future.

The important finding that the absence of acuK make this fungal susceptible is very interesting but author should discuss the therapeutic scope of these findings in the discussion section. What type of future studies should be conducted to progress this work for the translational aspects of these findings. 

We add more discussion on the therapeutic intervention of AcuK in the last paragraph of discussion (as appeared in revised manuscript, page 13-14), and two additional references were added (number 24, 25).
